# Quercetin Attenuates KLF4-Mediated Phenotypic Switch of VSMCs to Macrophage-like Cells in Atherosclerosis: A Critical Role for the JAK2/STAT3 Pathway

**DOI:** 10.3390/ijms25147755

**Published:** 2024-07-15

**Authors:** Lu Xiang, Yan Wang, Si Liu, Linyao Ying, Keyi Zhang, Na Liang, Hao Li, Gang Luo, Lin Xiao

**Affiliations:** Xiangya School of Public Health, Central South University, Changsha 410013, China; xianglu66@csu.edu.cn (L.X.); estherwyan@csu.edu.cn (Y.W.); 226911029@csu.edu.cn (S.L.); 226912059@csu.edu.cn (L.Y.); zhangky@csu.edu.cn (K.Z.); 236912047@csu.edu.cn (N.L.); 236912060@csu.edu.cn (H.L.)

**Keywords:** quercetin, atherosclerosis, macrophage-like VSMCs, phenotypic switch, JAK2/STAT3 pathway, KLF4

## Abstract

The objective of this study was to elucidate the protective role of quercetin in atherosclerosis by examining its effect on the phenotypic switch of vascular smooth muscle cells (VSMCs) to macrophage-like cells and the underlying regulatory pathways. Aorta tissues from apolipoprotein E-deficient (ApoE KO) mice fed a high-fat diet (HFD), treated with or without 100 mg/kg/day quercetin, were analyzed for histopathological changes and molecular mechanisms. Quercetin was found to decrease the size of atherosclerotic lesions and mitigate lipid accumulation induced by HFD. Fluorescence co-localization analysis revealed a higher presence of macrophage-like vascular smooth muscle cells (VSMCs) co-localizing with phospho-Janus kinase 2 (p-JAK2), phospho-signal transducer and activator of transcription 3 (p-STAT3), and Krüppel-like factor 4 (KLF4) in regions of foam cell aggregation within aortic plaques. However, this co-localization was reduced following treatment with quercetin. Quercetin treatment effectively inhibited the KLF4-mediated phenotypic switch in oxidized low-density lipoprotein (ox-LDL)-loaded mouse aortic vascular smooth muscle cells (MOVAS), as indicated by decreased expressions of KLF4, LGALS3, CD68, and F4/80, increased expression of alpha smooth muscle actin (α-SMA), reduced intracellular fluorescence Dil-ox-LDL uptake, and decreased lipid accumulation. In contrast, APTO-253, a KLF4 activator, was found to reverse the effects of quercetin. Furthermore, AG490, a JAK2 inhibitor, effectively counteracted the ox-LDL-induced JAK2/STAT3 pathway-dependent switch to a macrophage-like phenotype and lipid accumulation in MOVAS cells. These effects were significantly mitigated by quercetin but exacerbated by coumermycin A1, a JAK2 activator. Our research illustrates that quercetin inhibits the KLF4-mediated phenotypic switch of VSMCs to macrophage-like cells and reduces atherosclerosis by suppressing the JAK2/STAT3 pathway.

## 1. Introduction

Atherosclerosis, marked by plaque buildup in arteries, is the leading cause of cardiovascular disease-related illness and death worldwide [1]. This condition is a chronic inflammatory response triggered by the gradual accumulation of fats in the inner lining of arteries. Foam cells, primarily created by macrophages in artery walls as they ingest oxidized low-density lipoprotein (ox-LDL), are key elements of atherosclerotic plaques, with their quantity indicating the severity of atherosclerosis [2,3]. Regulating the transition of macrophages into foam cells is essential for preventing atherosclerosis.

Recently, increasing studies have highlighted the presence of foam cells derived from vascular smooth muscle cells (VSMCs) in atherosclerosis development [4]. VSMCs display strong phenotypic plasticity in atherosclerosis. VSMCs exhibit significant phenotypic flexibility in this process. The accumulation of lipids within plaques triggers the phenotypic switch of VSMCs into macrophage-like cells, characterized by the increased expression of typical macrophage markers (such as CD68, F4/80, and LGALS3) and the decreased expression of VSMC-specific markers (such as alpha smooth muscle actin, α-SMA) [5,6]. These VSMCs-derived macrophage-like cells exhibit heightened sensitivity to cholesterol levels and lipid loading, subsequently contributing to the formation of cholesterol-filled foam cells [7], emblematic pathological hallmarks of atherosclerosis. Allahverdian and colleagues [5] found that approximately 40% of all foam cells within human coronary atherosclerosis lesions come from VSMCs, not of leukocyte origin, as evidenced by co-staining with α-SMA and CD68. Wang and his team [8] discovered that approximately 60% to 70% of foam cells in atherosclerosis lesions in mice originated from VSMCs, as confirmed through SMC-lineage tracing. This suggests that controlling the phenotypic switch of vascular smooth muscle cells into macrophage-like cells could be a new strategy to reduce foam cell buildup and slow atherosclerosis progression. However, limited research has not yet fully elucidated the specific molecular mechanisms underlying this phenotypic switch.

Krüppel-like factor 4 (KLF4), functioning as a transcription factor, plays a critical role in regulating the phenotypic switch of vascular smooth muscle cells (VSMCs) to macrophage-like cells during the pathogenesis of atherosclerosis [9,10,11]. Shankman et al. [12] demonstrated that the specific deletion of *KLF4* in smooth muscle cells (SMCs) within apolipoprotein E-deficient (ApoE KO) mice significantly inhibited the phenotypic switch of VSMCs to macrophage-like cells, thereby reducing lesion size and enhancing plaque stability. Consequently, targeting the KLF4-mediated phenotypic modulation of VSMCs represents a promising therapeutic strategy for atherosclerosis.

The differentiation [13], proliferation [14], and migration [15] of VSMCs are closely linked to the activated Janus kinase 2 (JAK2)/signal transducer and activator of transcription 3 (STAT3) pathway. Recently, Wang et al. [16] demonstrated that urantide mitigates the phenotypic switch of VSMCs from a contractile to a synthetic state through the phosphorylation of the JAK2/STAT3 pathway, thereby attenuating atherosclerotic lesions in a high-fat diet (HFD)-fed rat model. Similarly, other studies have reported that the phosphorylation of the JAK2/STAT3 pathway is activated by platelet-derived growth factor BB homodimer or vascular endothelial growth factor, leading to a reduction in the expression of VSMC contractile genes such as α-SMA [17,18]. Notably, research by Niwa et al. [19] has notably demonstrated that activation of the JAK2/STAT3 pathway can upregulate KLF4 expression, thereby promoting the self-renewal of mouse embryonic stem cells [20]. However, the mechanisms by which the JAK2/STAT3 pathway may induce a KLF4-mediated phenotypic switch of VSMCs to macrophage-like cells within atherosclerotic plaques remain inadequately understood.

Given the pivotal role of this phenotypic switch in the pathogenesis of atherosclerosis, it is crucial to identify effective therapeutic agents that target the JAK2/STAT3/KLF4 pathway to inhibit the transition of VSMCs to macrophage-like cells. Quercetin, a naturally occurring flavonoid predominantly present in colorful vegetables and fruits, exhibits beneficial therapeutic effects on cardiovascular disease [21]. Evidence from both human [22] and animal studies [23,24] indicates that quercetin is associated with a significantly reduced risk of atherosclerosis. Current studies of substantial value demonstrate that the regulatory effects of quercetin on VSMCs primarily involve cellular senescence [25], proliferation, and migration [26]. Research conducted by Hu et al. [27] demonstrates that quercetin mitigates fructose-induced renal lipid accumulation in rats through the suppression of the JAK2/STAT3 pathway. However, no research has investigated the role of quercetin in phenotypic switch of VSMCs to macrophage-like cells during atherogenesis, and the underlying regulatory mechanisms remain unclear.

Building on these findings, our study examines the effects of quercetin on the phenotypic switch of VSMCs to macrophage-like cells and explores its potential molecular mechanisms in both ApoE KO mice and the mouse aortic vascular smooth muscle cell (MOVAS) line.

## 2. Results

### 2.1. Improvement of Dyslipidemia and Atherosclerosis in HFD-Fed ApoE KO Mice by Quercetin

To investigate the potential alleviating effects of quercetin on atherosclerosis, ApoE KO mice were either fed an HFD or supplemented with 100 mg/kg quercetin for 16 weeks. Figure 1A–D exhibited significantly elevated levels of serum total cholesterol (TC), triglycerides (TG), and low-density lipoprotein cholesterol (LDL-C), along with decreased levels of serum high-density lipoprotein cholesterol (HDL-C) in the HFD group compared to the normal chow diet (ND) group. Nevertheless, supplementation with quercetin markedly reduced serum TC, TG, and LDL-C levels while increasing HDL-C levels. These findings suggest that quercetin exerts beneficial effects in ameliorating dyslipidemia in ApoE KO mice.

Subsequently, the impact of quercetin on aortic plaques was assessed through H&E staining and Oil Red O staining. H&E staining (Figure 1E,G) revealed a significant deterioration in the plaque area at the aortic sinus in HFD-fed ApoE KO mice, which was effectively mitigated by quercetin treatment. Additionally, Oil Red O staining further demonstrated a marked reduction in lipid content in mice treated with quercetin under HFD conditions (Figure 1F,H). Based on these results, it can be concluded that quercetin effectively retards the development of HFD-induced atherosclerotic lesions.

### 2.2. Co-Localization of KLF4, p-JAK2, and p-STAT3 with Plaque Macrophage-like VSMCs in ApoE KO Mice

The macrophage-like VSMCs can concurrently express markers from two distinct lineages, such as α-SMA (labeling VSMCs) and CD68 (labeling macrophages). Previous studies have confirmed that KLF4 is a key regulatory protein for phenotypic switch of VSMCs to macrophage-like cells [12], and JAK2/STAT3 is considered as an upstream pathway of KLF4 [20]. To provide evidence for the roles of the JAK2/STAT3 pathway and KLF4 protein in regulating the phenotypic switch of VSMCs to macrophage-like cells and the intervention effect of quercetin, TSA multi-color immunofluorescence staining was employed to examine the co-localization of macrophage-like VSMCs, identified as double-positive for α-SMA (pink) and CD68 (green), with KLF4, p-JAK2, and p-STAT3 (red) in the aortic tissues of ApoE KO mice, respectively. The results revealed that, compared to the ND group, the HFD group exhibited conspicuous co-localization of α-SMA^+^CD68^+^KLF4^+^, α-SMA^+^CD68^+^p-JAK2^+^, and α-SMA^+^CD68^+^p-STAT3^+^ at the foam cell aggregation regions of atherosclerosis plaque-rich aortic. After quercetin intervention, there was a notable reduction in the range of fluorescence co-localization, as depicted in Figure 2. These findings suggest the potential constructive roles of the JAK2/STAT3 pathway and KLF4 protein in facilitating the phenotypic switch of VSMCs to macrophage-like cells within atherosclerosis plaques and indicate a potential mechanism through which quercetin exerts its protective effects against atherosclerosis.

### 2.3. Ox-LDL Treatment Triggered Phenotypic Switch of VSMCs to Macrophage-like Cells in MOVAS

To understand the effect of ox-LDL on the phenotypic switch of VSMCs to macrophage-like cells in vitro, MOVAS were treated with various concentrations of ox-LDL alone for 24 h. Initially, the CCK-8 assay (Figure 3A) demonstrated that exposure to ≤100 µg/mL ox-LDL was significantly non-cytotoxic to viability of MOVAS. Additionally, Oil Red O staining (Figure 3B) revealed a dose-dependent augmentation in foam cell formation with escalating ox-LDL concentrations. Subsequently, to further explore the association between ox-LDL-induced lipid accumulation and cholesterol uptake, MOAVS were incubated with fluorescence Dil-ox-LDL for 4 h following pretreatment with ox-LDL. The results illustrated a dose-dependent enhancement in the ability of MOVAS to internalize cholesterol upon exposure to ox-LDL (Figure 3C). Moreover, Western blot analysis (Figure 3D–H) confirmed that exposure of MOVAS to 100 µg/mL ox-LDL noticeably upregulated the expression of macrophage markers F4/80, CD68, and LGALS3, while concurrently downregulating the expression of VSMCs contraction marker α-SMA (all *p* < 0.01). Thus, 100 µg/mL ox-LDL was identified as the optimal concentration for inducing phenotypic switch of MOVAS to macrophage-like cells in vitro, consequently enhancing cholesterol uptake capacity and culminating in foam cell formation.

### 2.4. Quercetin Suppressed Foam Cell Formation by Inhibition of Phenotypic Switch of VSMCs to Macrophage-like Cells

MOVAS were exposed to 20 μmol/L quercetin for 24 h in the presence of 100 µg/mL ox-LDL to verify its inhibitory effect on ox-LDL-induced phenotypic switch of VSMCs to macrophage-like cells. The CCK-8 assays (Figure 4A) revealed that below the concentration of 20 µmol/L, quercetin had no significant changes in the viability of MOVAS. Figure 4B demonstrates that cellular lipid accumulation in MOVAS increased after ox-LDL stimulation, while quercetin supplementation effectively reduced this rise in lipid content. Additionally, Dil-ox-LDL uptake results indicated that quercetin significantly impeded the cholesterol uptake process in MOVAS (Figure 4C). These findings suggest that quercetin has the potential to partly inhibit cholesterol uptake, thereby hindering foam cell formation in MOVAS. Moreover, compared to the ox-LDL group, quercetin supplementation markedly reversed the ox-LDL-induced levels of F4/80, CD68, and LGALS3 and counteracted the ox-LDL-induced reduction in α-SMA levels, further illustrating its inhibitory effect on the phenotypic switch of VSMCs to macrophage-like cells through Western blot analysis (Figure 4D–H). These results support the capacity of quercetin to reverse the ox-LDL-induced phenotypic switch of VSMCs to macrophage-like cells. Summarily, inhibition of cholesterol uptake and foam cell formation by quercetin was through preventing phenotypic switch of VSMCs to macrophage-like cells.

### 2.5. Quercetin Blocked Phenotypic Switch of VSMCs to Macrophage-like Cells and Foam Cell Formation through Reducing KLF4 Expression

As shown in Figure 5A,B, we observed that KLF4 protein expression increased after ox-LDL stimulation, as anticipated, but decreased with quercetin treatment. To further confirm whether KLF4 is a crucial regulator of quercetin’s inhibition of phenotypic switch of VSMCs to macrophage-like cells, we stimulated MOVAS exposed to 100 µg/mL ox-LDL with 20 µmol/L quercetin in vitro, followed with 5 µmol/L APTO-253, a specific inducer of KLF4. The APTO-253 dosage in this study was based on a previous study [28] and cell viability assay results. Appendix A demonstrated that 5 μmol/L of APTO-253 had no observable impact on MOVAS viability. As mentioned earlier, Oil Red O staining and Dil-ox-LDL uptake assay indicated that the APTO-253 reversed the inhibitory effects of quercetin on lipid accumulation (Figure 5C) and cholesterol uptake (Figure 5D) induced by ox-LDL. Western blot analysis (Figure 5E–J) consistently showed that quercetin suppressed the KLF4 level and phenotypic switch of VSMCs to macrophage-like cells, while APTO-253 reversed these effects, implying that quercetin acts by decreasing KLF4 level. These results suggest that quercetin modulates KLF4 expression to inhibit phenotypic switch of VSMCs to macrophage-like cells, thereby preventing lipid deposition.

### 2.6. Ox-LDL Induced Phenotypic Switch of VSMCs to Macrophage-like Cells in MOVAS by Activating JAK2/STAT3 Pathway

To directly confirm whether ox-LDL can trigger KLF4-mediated phenotypic switch of VSMCs to macrophage-like cells via the JAK2/STAT3 pathway, MOVAS were treated with 25 µmol/L AG490, a specific inhibitor of JAK2/STAT3, in the presence of 100 μg/mL ox-LDL. The concentration of AG490 was chosen based on a previous study [29] and the CCK-8 assay result (Appendix A). As illustrated in Figure 6A–D, AG490 treatment led to a reduction in the number of foam cells derived from VSMCs compared to the ox-LDL group, accompanied by a decrease in lipid uptake induced by ox-LDL stimulation. Furthermore, AG490 downregulated the phosphorylation levels of JAK2 and STAT3, as well as the expression of KLF4. Additionally, it inhibited the phenotypic switch of VSMCs to macrophage-like cells, characterized by an increase in α-SMA expression and a decrease in the expression of F4/80, CD68, and LGALS3 (Figure 6E–L). These findings strongly indicate that ox-LDL promotes the phenotypic switch of VSMCs to macrophage-like cells by activating the JAK2/STAT3 pathway.

### 2.7. Quercetin Improved the ox-LDL-Induced Phenotypic Switch of VSMCs to Macrophage-like Cells in MOVAS by Reduction of the Activated JAK2/STAT3 Pathway

We next investigated whether quercetin’s protective effect against ox-LDL-induced phenotypic switch of VSMCs to macrophage-like cells is mediated through the JAK2/STAT3 pathway. MOVAS treated with 100 μg/mL ox-LDL were loaded with either 20 µmol/L quercetin or 10 µmol/L coumermycin A1 (CA1). The concentration of CA1 was chosen according to a prior study [30] and the CCK-8 assay result, as depicted in Appendix A. Compared with the ox-LDL+ quercetin group, treatment with CA1 increased lipid accumulation (Figure 7A) and abrogated the reduction of cholesterol uptake ability (Figure 7B) in cells incubated with quercetin. Additionally, CA1 supplementation reversed the decrease in levels of KLF4, F4/80, CD68, and LGALS3, as well as the increase in α-SMA levels, while suppressing the JAK2 and STAT3 expressions at phosphorylation levels (Figure 7C–J). Our data indicate that the protective effect of quercetin on the phenotypic switch of VSMCs to macrophage-like cells is achieved by reducing JAK2/STST3 pathway phosphorylation, consequently decreasing VSMC-derived foam cells.

## 3. Discussion

The phenotypic switch of VSMCs to macrophage-like cells is increasingly recognized as a primary contributor to foam cell formation within atherosclerotic plaques [8]. However, limited research has not fully elucidated the underlying molecular mechanisms and promising therapeutic targets for this complex pathological process. This study investigated the beneficial effects of the flavonoid compound quercetin on atherosclerosis, specifically focusing on VSMCs phenotype switch process. Our findings provide, for the first time, novel evidence that quercetin suppresses the phenotypic switch of VSMCs to macrophage-like cells by inhibiting the JAK2/STAT3 pathway and reducing KLF4 expression, thereby preventing foam cell formation and attenuating atherosclerosis.

An increasing number of studies have demonstrated the protective effects of quercetin against atherosclerosis by regulating lipid levels and preventing TC accumulation [22,23,24]. In the present study, we observed that chronic quercetin treatment effectively improved HFD-induced dyslipidemia in ApoE KO mice, with a significant reduction in TC, TG, and LDL-C levels, as well as an increase in HDL-C levels. Most importantly, quercetin significantly attenuated the atherosclerotic lesions by decreasing the lipid deposition and plaque adhesion in the aorta, which is consistent with previous studies [24]. Our in vivo results suggest that quercetin may hinder the advancement of atherosclerosis by mitigating hypercholesterolemia and preventing the phenotypic switch of VSMCs into macrophage-like cells. Nonetheless, additional research is required to elucidate the primary mechanisms through which quercetin exerts its atheroprotective effects.

There is increasing evidence that macrophage-like VSMCs promote the development of atherosclerosis by contributing to the formation of cholesterol-filled foam cells [7,30]. Ceccherini et al. [31] found that quercetin counteracted vascular calcification (a predictor of atherosclerotic vascular disease mortality) in primary HCASMCs by attenuating an osteoblast-like phenotypic switch. However, current studies targeting the phenotypic switch of VSMCs to macrophage-like cells in delaying atherosclerosis are limited. In our study, obvious macrophage-like VSMCs were reduced by quercetin in ox-LDL-induced MOVAS, as indicated by the decreased macrophage markers (CD68, F4/80, and LGALS3) and the increased α-SMA levels. Additionally, besides inhibiting the phenotypic switch of VSMCs to macrophage-like cells, the numbers of ox-LDL-induced foam cells were significantly decreased by quercetin. Foam cell formation is evidently caused by an imbalance of intracellular lipid homeostasis between cholesterol uptake and cholesterol efflux. Existing research has shown that quercetin can improve lipid metabolism disorders in atherosclerosis by inhibiting lipid uptake [32]. Consistent with this finding, our study observed that quercetin reduced cholesterol uptake by inhibiting the phenotypic switch of VSMCs to macrophage-like cells, thereby reducing foam cell numbers and delaying the progression of atherosclerosis. The effect of quercetin on regulating lipid metabolism involves downregulating expression of the scavenger receptor CD36 and upregulating expression of the cholesterol efflux regulatory gene ATP-binding cassette transporter A1 (ABCA1) expression [33]. Foam cells derived from VSMCs express lower levels of the ABCA1, resulting in ineffective cholesterol expulsion [34]. Therefore, regulating the expression of scavenger receptor or cholesterol efflux regulatory protein for the alleviating cholesterol burden of macrophage-like VSMCs may represent a potential option for quercetin in delaying the progression of atherosclerosis.

The JAK2/STAT3 pathway was significantly upregulated in HFD-fed ApoE KO mice [33], and blocking the JAK2/STAT3 pathway can prevent the formation of atherosclerotic lesions [35,36]. Furthermore, KLF4, a key regulator of VSMCs phenotypic switch, is recognized as a downstream target of STAT3 [19]. In this study, our multi-color immunofluorescence results indicated increased co-localization of KLF4, p-JAK2, and p-STAT3 with macrophage-like VSMCs (α-SMA^+^CD68^+^ double positive cells) in plaques with dense foam cell aggregation in HFD-treated mice. Treatment with the JAK2/STAT3 inhibitor AG490 abolished the effects of ox-LDL on KLF4-mediated phenotypic switch to macrophage-like cells in MOVAS. Consistent with our results, previous studies have reported that ox-LDL triggers VSMCs phenotypic switch by activating JAK2/STAT3 pathway [16,19]. Furthermore, Xue et al. [37] reported that the expression of p-STAT3 peaked in macrophage-like VSMCs to promote atherosclerosis in HFD-fed ApoE KO mice at the early stage, and progressively downregulated in response to macrophage-derived crosstalk, indicating dynamic changes in the function of macrophage-like VSMCs during atherosclerosis.

Although quercetin exhibits anti-atherosclerotic properties [23,24], whether it can ameliorate ox-LDL-induced phenotypic switch of VSMCs to macrophage-like cells within plaques and mitigate atherosclerosis remains uncertain. Given that quercetin has been reported to reverse fructose-induced renal lipid accumulation in rats by suppressing JAK2/STAT3 [27], it is speculated that quercetin delays ox-LDL-induced phenotypic switch of VSMCs to macrophage-like cells in plaques by inhibiting the JAK2/STAT3 pathway. Our study found that intervention with quercetin in HFD-treated mice reduced the immunofluorescence co-localization range of p-JAK2 and p-STAT3 with macrophage-like VSMCs, leading to a reduction in plaque lesions. Moreover, quercetin suppressed ox-LDL-induced KLF4-mediated phenotypic switch to macrophage-like cells in MOVAS and inhibited the phosphorylation of JAK2/STAT3, and the JAK2/STAT3 activator CA1 reversed the effects of quercetin in MOVAS. Consistent with these results, research from Wang et al. [38] revealed that quercetin selectively inhibited the phosphorylation of JAK2/STAT3 in VSMCs, thereby attenuating the angiotensin II-stimulated cell proliferation. Another study from Xi et al. [39] also demonstrated that quercetin alleviates the oxidative damage by suppressing the KLF4 expression in neuroblastoma SH-SY5Y cells.

To our knowledge, this is the first time that quercetin has been used as a therapeutic strategy to inhibit KLF4-mediated atherosclerosis through the JAK2/STAT3 pathway. Recent research has shown that the transcription factor homeobox A1 participates in atherosclerosis progression by activating KLF4 expression, thereby promoting the phenotypic switch of VSMCs to macrophage-like cells [10]. Similarly, Vendrov et al. [40] found that NOX activator 1-dependent NADPH oxidase promotes the phenotypic switch of VSMCs to macrophage like cells and plaque inflammation in atherosclerosis by upregulating KLF4 expression. Intriguingly, another study found that formononetin reduces scavenger receptor class A expression by regulating the expression and nuclear translocation of KLF4, which substantially attenuated VSMCs-derived foam cell accumulation [11]. In fact, it is noteworthy that the promoting or inhibitory effects of KLF4 in atherosclerosis primarily depend on the specific target cells. Recent research has revealed that C1q/tumor necrosis factor-related protein 9 relies on the AMPKα pathway to reduce KLF4 expression, inhibit hyperglycemia-induced endothelial senescence, and attenuate atherosclerosis [41]. However, high-level shear stress-activated KLF4 expression can alleviate vascular endothelial cell inflammation and thus produce anti-atherosclerotic effects [42]. Therefore, further investigation is needed to determine the precise role of quercetin in preventing atherosclerosis through its interaction with KLF4. 

## 4. Materials and Methods

### 4.1. Chemical Reagents and Antibodies

All chemical reagents and antibodies in this study were used following the provided instructions. Quercetin with 98% purity was obtained from Sigma-Aldrich (St. Louis, MO, USA), and ox-LDL was provided by Yiyuan Biotechnology (Guangzhou, China). APTO-253, the KLF4 activator, was purchased from GlpBiO (Montclair, CA, USA); AG490, the JAK2 inhibitor, was procured from MCE (San Rafael, CA, USA); and coumermycin A1, the JAK2 activator, was purchased from Promega (Madison, WI, USA). The following primary antibodies were obtained: JAK2, STAT3, p-STAT3, CD68, F4/80 and GAPDH (CST, Danvers, MA, USA), KLF4 and α-SMA (Novus Biologicals, Centennial, CO, USA), LGALS3 (Santa Cruz Biotechnology, Dallas, TX, USA), and p-JAK2 (Abcam, Cambridge, UK). The secondary antibodies were goat anti-rabbit IgG-HRP and goat anti-mouse IgG-HRP obtained from CST (Danvers, MA, USA). Tyramide signal amplification (TSA) multiplex immunofluorescence kit was provided by AiFang Biotechnology (Hunan, China).

### 4.2. Animal Experiments

The procedure’s in vivo studies were performed in line with the Experimental Animal Ethics Committee of Central South University. Male ApoE KO mice (six weeks old) with C57BL/6J background (n = 30) were purchased from Beijing Vital River Laboratory Animal Technology Co., Ltd. (Beijing, China) and housed in a controlled environment (temperature: 22 ± 2 °C; humidness: 50 ± 5%; 12-h light/dark cycle). After 1 week of adaptive feeding prior to experiments, ApoE KO mice were randomly assigned to the following groups (10 mice per group): ND group; HFD group, consisted of 1.25% cholesterol and 21% fat; HFD + Que group, received HFD supplemented with 100 mg/kg/day quercetin. All mice received saline or quercetin (intragastric administration) daily for 16 weeks. Following overnight fasting and euthanasia, aortas and blood samples from all ApoE KO mice were collected for subsequent analyses.

### 4.3. Atherosclerosis Lesion Assessment and Immunofluorescence

As previously described [24], the aortic tissues were fixed in 4% paraformaldehyde and embedded in OCT compound, followed being sliced into continuous transverse sections (each 5 μm-thick) for the evaluation of atherosclerotic lesions. H&E staining and Oil Red O (Servicebio, Wuhan, China) staining were conducted to examine the area of atherosclerotic lesions and lipid deposition, monitoring staining intensity under an optical microscope.

For immunofluorescence staining, aortic tissue sections were incubated with corresponding primary antibodies (anti-CD68, CST97778, 1:500; anti-α-SMA, NBP2-33006, 1:500; anti-KLF4, NBP2-24749, 1:200; anti-p-STAT3, CST9145, 1:400; anti-p-JAK2, ab32101, 1:200) overnight, and revealed by secondary antibodies. The sections were then reacted with TYR-520 (green), TYR-570 (red), and TYR-690 (pink) fluorescent reagents of the TSA reagent kit (AFIHC024), respectively. Furthermore, DAPI was employed as a nuclear counterstain. All experiments were carried out in accordance with the manufacturer’s instructions, and fluorescent images were obtained by the Zeiss LSM 510 laser scanning confocal microscopes.

### 4.4. Biochemical Measurements

Following the guidelines provided by the manufacturer, serum samples were processed to measured TG, TC, LDL-C, and HDL-C levels using assay kits (Bioengineering Institute of Nanjing Jiancheng, Nanjing, China).

### 4.5. MOVAS Culture

MOVAS was obtained from the ATCC (CRL-2797, Manassas, VA, USA) and routinely cultured in high-glucose DMEM (Titan Scientific, Shanghai, China) supplemented with 10% FBS (GIBCO, Vacaville, CA, USA) and 1% penicillin/streptomycin (Titan Scientific, Shanghai, China) at 37 °C in a humidified incubator containing 5% CO_2_. MOVAS in the logarithmic growth phase were selected and seeded in cell culture dishes for the respective experiments. According to the cell viability results and previous studies [28,29,30], cells were stimulated by selected compounds, including 100 µg/mL ox-LDL, 20 μmol/L quercetin, 25 μmol/L AG490, 5 μmol/L APTO-253, or 10 μmol/L CA1, without cytotoxicity to MOVAS in the proper dosages. After 24 h, cells were used for subsequent experiments.

### 4.6. Cell Viability Assay

MOVAS were evenly distributed into 96-well plates and incubated overnight. Following treatment with various concentrations of ox-LDL (0, 25, 50, 75, or 100 µg/mL), quercetin (0, 5, 10, 20, 40, 80, or 160 µmol/L), AG490 (0, 5, 10, 25, 50, or 100 µmol/L), APTO-253 (0, 1, 5, 10, 20, or 50 µmol/L), or Coumermycin A1 (0, 1, 5, 10, 20, or 50 µmol/L) for 24 h, cell viability was assessed utilizing the CCK-8 assay (APExBIO, Boston, MA, USA).

### 4.7. Western Blot

Protein extraction from treated cells was carried out using RIPA lysis buffer encompassing the phosphatase inhibitor, followed by quantification of protein concentration using the bicinchoninic acid method. Protein samples were separated by 10% SDS-PAGE and then transferred to 0.45 μm PVDF membrane (Millipore, Burlington, MA, USA). After blocking, the membrane was incubated with particular primary antibodies (JAK2, 1:1000; p-JAK2, 1:2000; STAT3, 1:1000; p-STAT3, 1:2000; KLF4, 1:1000; F4/80, 1:1000; CD68, 1:1000; α-SMA, 1:1000; LGALS3, 1:1000; GAPDH, 1:4000) overnight at 4 °C. Subsequently, the bands were incubated with respective secondary antibodies for 1 h and visualized via ECL (Share-bio Biotechnology, Shanghai, China). The band densities were quantified by Image J 1.8.0 and normalized to the GAPDH level.

### 4.8. Oil Red O Staining

The deposition of cellular lipids in MOVAS was evaluated through Oil Red O staining. As described previously [24], cells with different treatments were fixed in 4% ice-cold paraformaldehyde for 1 h, followed by three washes with PBS and subsequent staining with 0.5% Oil Red O. Images were visualized under an optical microscope and the positive staining area was quantified with Image J Pro Plus 6.0 software.

### 4.9. Dil-ox-LDL Uptake Analysis

Following various treatments, cells were incubated with 10 μg/mL fluorescence 1,1′-dioctadecyl-3,3,3′,3′-tetramethyl-indocarbocyanine-ox-LDL (Dil-ox-LDL) (Yiyuan Biotechnology, Guangzhou, China) for 4 h, as described earlier [34]. Washing with PBS three times to remove surface adherent Dil-ox-LDL, cells were imaged by a fluorescence microscope to measure the uptake level of lipids.

### 4.10. Statistical Analysis

All analyses were conducted using SPSS 18.0. Quantitative data of all repeated experiments were reported as mean ± SD. Statistical comparisons among multiple groups were determined using Student’s *t*-test or one-way ANOVA followed by LSD post hoc test. *p* < 0.05 indicates statistical significance.

## 5. Conclusions

Our study provides the first confirmation that quercetin inhibits the progression of atherosclerosis by regulating VSMCs’ phenotypic plasticity. Specifically, this protective effect of quercetin against atherosclerosis is partly attributed to its inhibition of phenotypic switch of VSMCs to macrophage-like cells and subsequent foam cell formation through a JAK2/STAT3/KLF4-dependent mechanism, providing a novel approach for attenuating atherosclerosis.

## Figures and Tables

**Figure 1 ijms-25-07755-f001:**
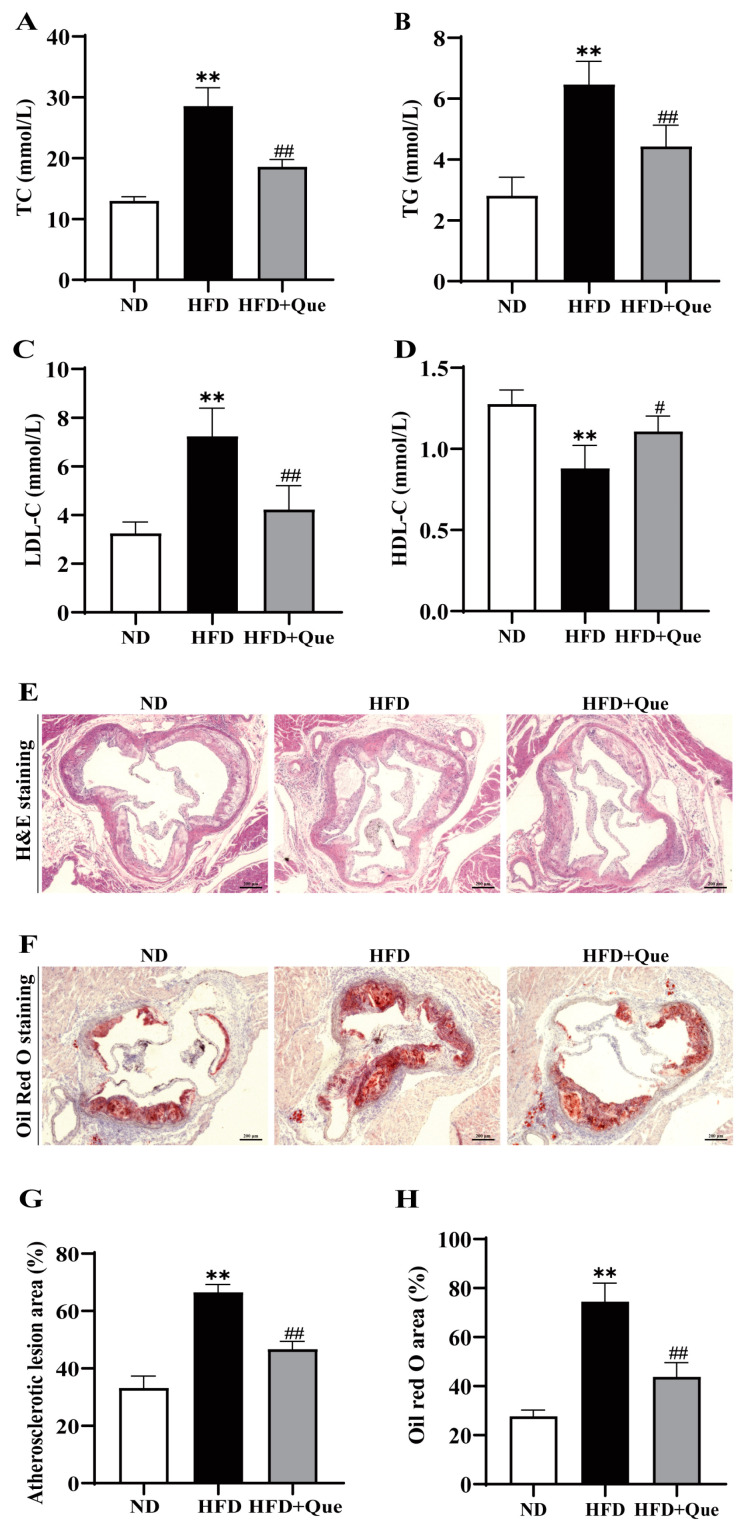
Improvement of dyslipidemia and atherosclerosis in high-fat diet (HFD)-fed apolipoprotein E-deficient (ApoE KO) mice by quercetin. (**A**–**D**) The serum total cholesterol (TC) (**A**), triglycerides (TG) (**B**), low-density lipoprotein cholesterol (LDL-C) (**C**), and high-density lipoprotein cholesterol (HDL-C) (**D**) levels. n = 7. (**E**–**H**) Representative images and quantification of aortic cross-sections stained with H&E and Oil Red O staining. Scale bar = 200 μm, 40×; n = 3. ** *p* < 0.01 vs. normal chow diet (ND) group; ^#^ *p* < 0.05, ^##^ *p* < 0.01 vs. HFD group.

**Figure 2 ijms-25-07755-f002:**
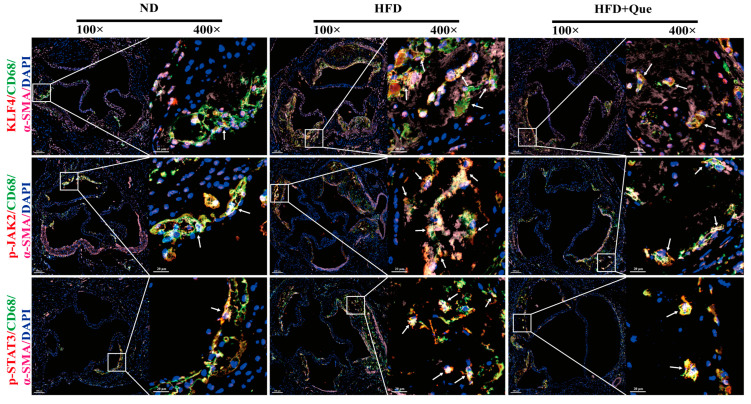
Representative immunofluorescence images of aortic cross-sections to illustrate the co-localization of α-SMA (pink), CD68 (green) with KLF4, p-JAK2, and p-STAT3 (red). Arrowheads indicate positive staining areas. Nuclei were stained with DAPI (blue). Scale bar =100 μm, 100×; scale bar = 20 μm, 400×; n = 3.

**Figure 3 ijms-25-07755-f003:**
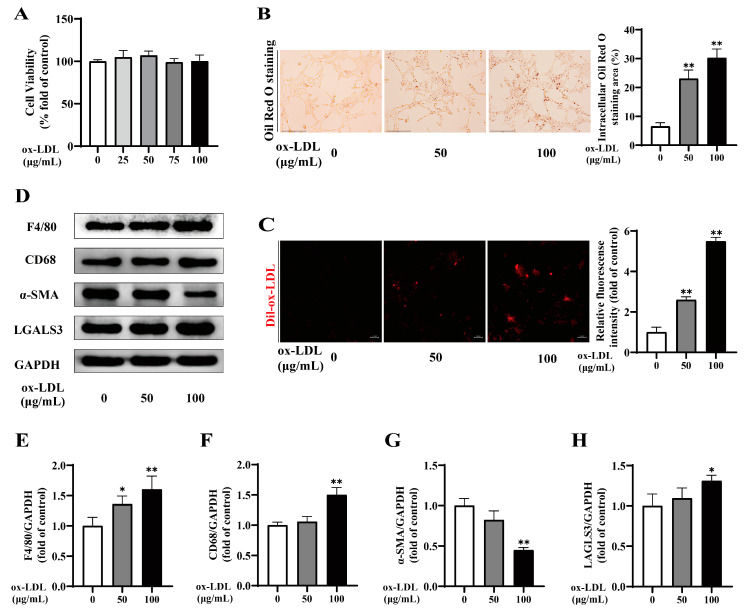
Oxidized low-density lipoprotein (ox-LDL) treatment triggered phenotypic switch of vascular smooth muscle cells (VSMCs) to macrophage-like cells in MOVAS at different concentrations. (**A**) Evaluation of MOVAS cell viability upon exposure to various concentrations of ox-LDL using the CCK8 assay, n = 6. (**B**) Representative images (left, scale bar = 150 µm, 200×) and quantitative analysis (right) of Oil Red O staining in ox-LDL-treated MOVAS. (**C**) Representative fluorescence images (left, scale bar = 20 µm) and quantitative analysis (right) of MOVAS treated with ox-LDL and labeled with Dil-ox-LDL. (**D**–**H**) Representative western blot and quantitative analysis for F4/80, CD68, alpha smooth muscle actin (α-SMA), LGALS3, and GAPDH proteins in MOVAS. * *p* < 0.05, ** *p* < 0.01 vs. 0 µg/mL ox-LDL group.

**Figure 4 ijms-25-07755-f004:**
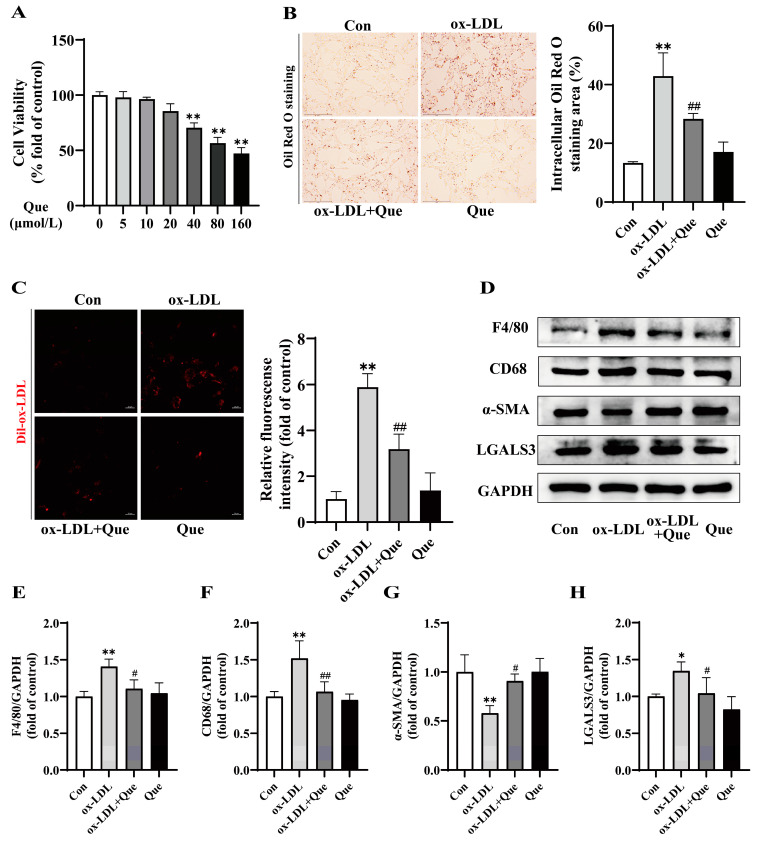
Quercetin suppressed foam cell formation by inhibition of phenotypic switch of VSMCs to macrophage-like cells. (**A**) Evaluation of MOVAS cell viability upon exposure to various concentrations of quercetin using the CCK-8 assay, n = 6. (**B**) Representative images (left, scale bar = 150 µm, 200×) and quantitative analysis (right) of Oil Red O staining in MOVAS. (**C**) Representative fluorescence images (left, scale bar = 20 µm) and quantitative analysis (right) of MOVAS labeled with Dil-ox-LDL. (**D**–**H**) Representative Western blot and quantitative analysis for F4/80, CD68, α-SMA, LGALS3, and GAPDH proteins in MOVAS. * *p* < 0.05, ** *p* < 0.01 vs. control group; ^#^
*p* < 0.05, ^##^
*p* < 0.01 vs. ox-LDL group.

**Figure 5 ijms-25-07755-f005:**
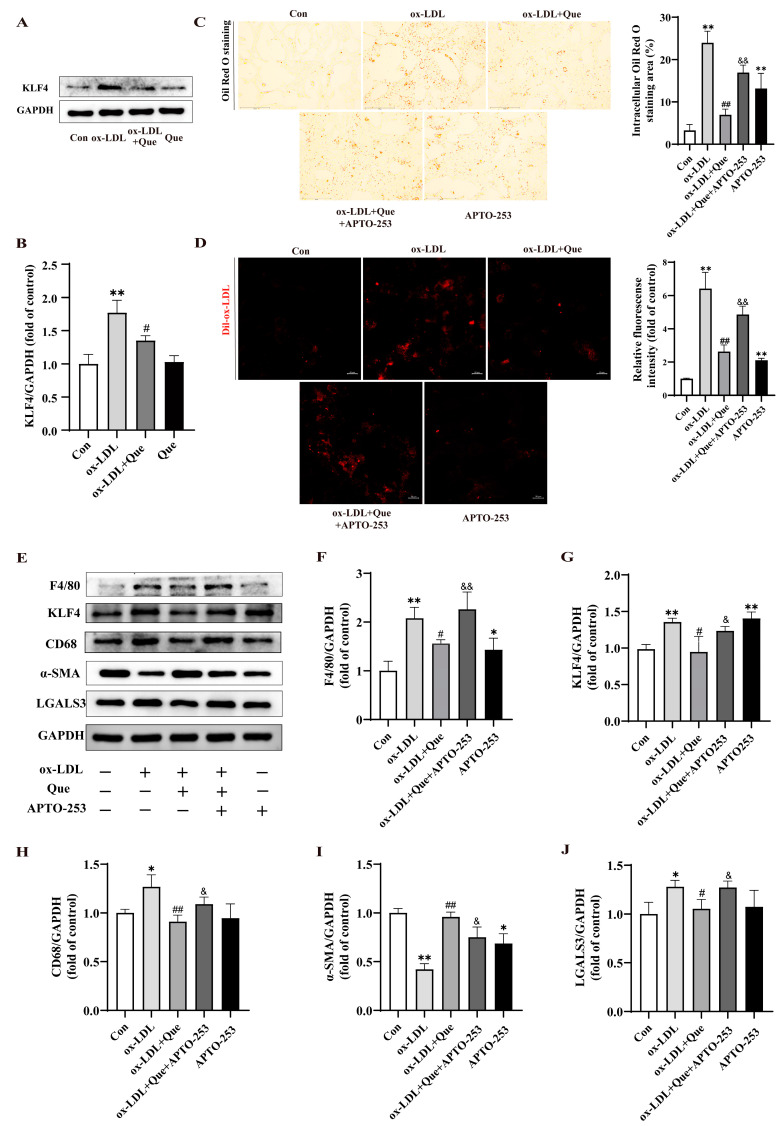
Quercetin blocked phenotypic switch of VSMCs to macrophage-like cells and foam cell formation through reducing Krüppel-like factor 4 (KLF4) expression. (**A**) Representative Western blot for KLF4 protein in MOVAS. (**B**) Quantitative analysis for the relative protein expression of KLF4 in MOVAS. (**C**) Representative images (left, scale bar = 150 µm, 200×) and quantitative analysis (right) of Oil Red O staining in MOVAS. (**D**) Representative fluorescence images (left, scale bar = 20 µm) and quantitative analysis (right) of MOVAS labeled with Dil-ox-LDL. (**E**–**J**) Representative Western blot and quantitative analysis for F4/80, KLF4, CD68, α-SMA, LGALS3, and GAPDH protein in MOVAS * *p* < 0.05, ** *p* < 0.01 vs. control group; ^#^ *p* < 0.05, ^##^ *p* < 0.01 vs. ox-LDL group; ^&^ *p* < 0.05, ^&&^ *p* < 0.01 vs. ox-LDL + Que group.

**Figure 6 ijms-25-07755-f006:**
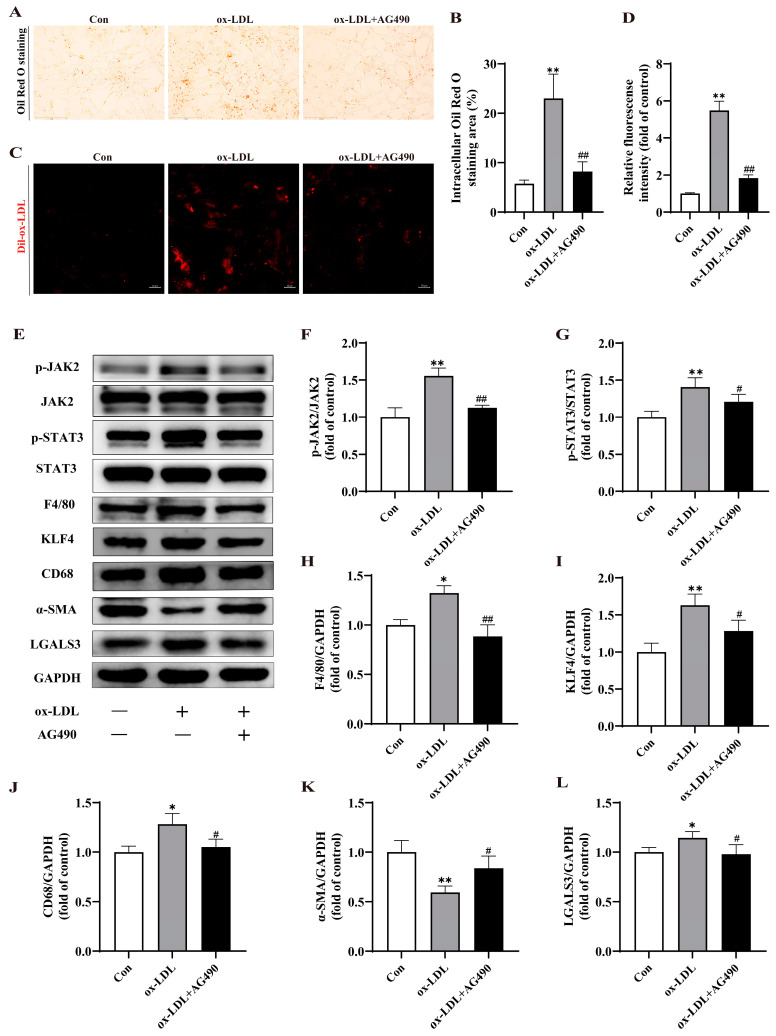
Ox-LDL-induced phenotypic switch of VSMCs to macrophage-like cells in MOVAS by activating Janus kinase 2 (JAK2)/signal transducer and activator of transcription 3 (STAT3) pathway. (**A**,**B**) Representative images (**A**) (scale bar = 150 µm, 200×) and quantitative analysis (**B**) of Oil Red O staining in MOVAS. (**C**,**D**) Representative fluorescence images (**C**) (scale bar = 20 µm) and quantitative analysis (**D**) of MOVAS labeled with Dil-ox-LDL. (**E**–**L**) Representative Western blot and quantitative analysis for p-JAK2, JAK2, p-STAT3, STAT3, F4/80, KLF4, CD68, α-SMA, LGALS3, and GAPDH protein in MOVAS. * *p* < 0.05, ** *p* < 0.01 vs. control group; ^#^ *p* < 0.05, ^##^ *p* < 0.01 vs. ox-LDL group.

**Figure 7 ijms-25-07755-f007:**
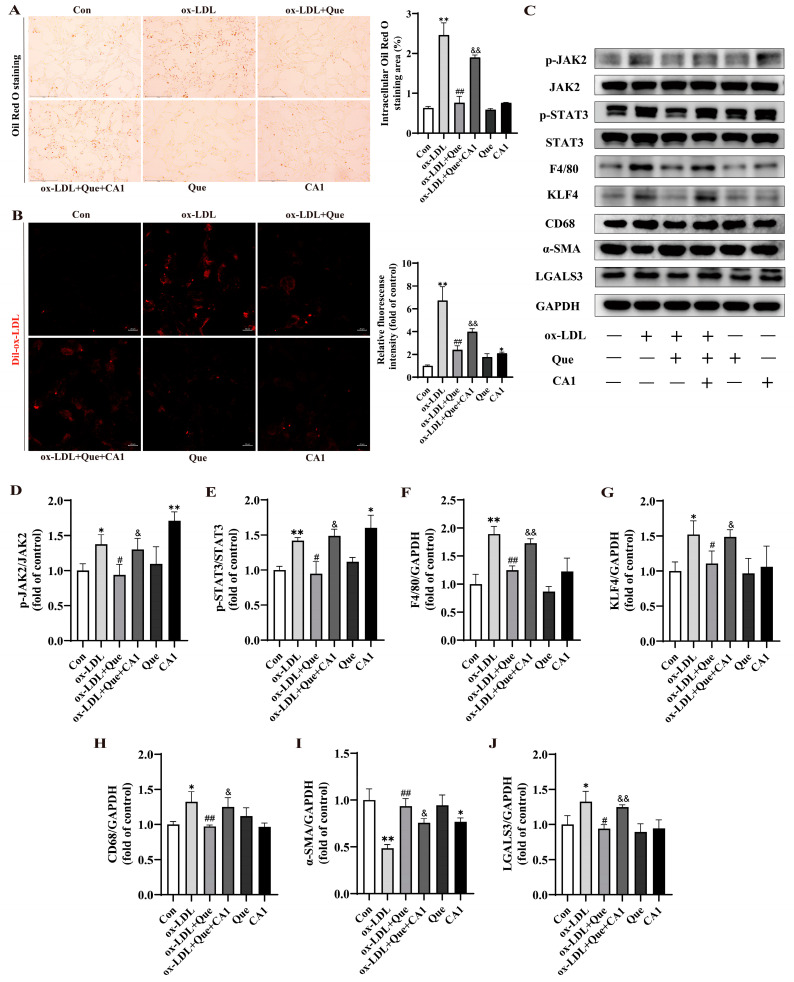
Quercetin improved the ox-LDL-induced phenotypic switch of VSMCs to macrophage-like cells in MOVAS by reduction of the activated JAK2/STAT3 pathway. (**A**) Representative images (left, scale bar = 150 µm, 200×) and quantitative analysis (right) of Oil Red O staining in MOVAS. (**B**) Representative fluorescence images (left, scale bar = 20 µm) and quantitative analysis (right) of MOVAS labeled with Dil-ox-LDL. (**C**–**J**) Representative Western blot and quantitative analysis for p-JAK2, JAK2, p-STAT3, STAT3, F4/80, KLF4, CD68, α-SMA, LGALS3, and GAPDH protein in MOVAS. * *p* < 0.05, ** *p* < 0.01 vs. control group; ^#^ *p* < 0.05, ^##^ *p* < 0.01 vs. ox-LDL group; ^&^ *p* < 0.05, ^&&^ *p* < 0.01 vs. ox-LDL + Que group.

## Data Availability

Data are contained within the article.

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
