# Peer review of "Quercetin Attenuates KLF4-Mediated Phenotypic Switch of VSMCs to Macrophage-like Cells in Atherosclerosis: A Critical Role for the JAK2/STAT3 Pathway"

_ijms, 2024, doi:10.3390/ijms25147755_

Round 1

Reviewer 1 Report

Comments and Suggestions for Authors

Xiang et al. examined the effect of quercetin on phenotypic switching of smooth muscle cells (SMCs) to macrophage-like cells in atherosclerosis.  They showed that quercetin prevented KLF4-mediated phenotypic switch of SMCs to macrophage-like cells via inhibiting the JAK2/STAT3 pathway.  The following points should be addressed to improve the manuscript. 

Comments:

1. The authors showed that quercetin decreased atherosclerosis in mice.  However, it is unclear whether this effect was mediated by quercetin-induced prevention of SMC phenotypic switching or by improving hypercholesteremia. 

2. Immunofluorescence studies are unclear in mice (Figure 2).  For example, SM a-actin should be stained in the cytoplasm, but it seems to be stained in the nuclei. 

3. In Figure 3D, F4/80 and CD68 were expressed abundantly in SMCs.  Did the authors check the contamination of macrophages in cultured SMCs?  Immunofluorescence studies are also required in cultured SMCs. 

4. Most mechanical studies were performed only in cultured SMCs.  Some of these studies should be examined in mice in vivo to confirm the results. 

5. “To understand the inhibitory effect of ox-LDL on….” on page 6, line 150.  This statement is not correct. 

6. English writing should be improved. 

Comments on the Quality of English Language

6. English writing should be improved. 

Reviewer 2 Report

Comments and Suggestions for Authors

General comment: The scientific work presented demonstrates the interesting versatility of atherosclerosis and reveals a fundamental signaling pathway for the switch of VSMCS in Macrofagi-like. In genreale the work is well presented and designed, my advice is to minimize repetition between the various sections: introduction -discussion or materials and methods- results.

I suggest to the authors to evaluate, in this work or in the future, how much this phenotypic switch and the attenuating action of KLF4 is mediated by the consilidata antioxidant action of quercetin, and therefore mediated also by other signaling or the action is preponderant on KLF4. 

In the introduction I suggest to better describe the macrophages-like phenotype and what characteristics/markers define it.

in each section of the manuscript, please reduce the repetitions, concepts and details already described in other paragraphs (for example: line 124-128, 150-159 and many more).

Results: Review the statistics so that the ranges of significance are consistent between the different graphs (for example: ** does not always correspond to all the differences between control and bars considered).

It is unclear why, if the control in the figures has value 1, and then all values are normalized on them (using the control as a reference), there is a standard deviation on the control bar. Please clarify this point in the captions or in the materials and methods.

line 134-136: rewrite more clearly.

Arrows may help in the images to indicate the highlights to look at, especially in IF on fabric.

Figures: In general, I suggest that authors increase the magnification or intensity of the Dil-ox-LDL images so that it is more visible and appreciable by the reader. For Oil Red O figures, I suggest increasing the contrast so that the differences are more visible.

In the panels containing the blots, I recommend replacing the blots too impressed and over exposed, in fact too intense and coarse bands interfere with the quantification, also preventing to clearly observe the modulation of the analyzed markers.

Line 189-190: "strongly reversed" is not observable by the blots presented. Please clarify this discrepancy.

Discussion: The discussion is well written, my concern is about the repetition of concepts that overlap with the introduction. 

 Materials and methods: Please enter the stimulation times of in vitro experiments to evaluate the protein expression of the presented markers.

Author Response

Title: Quercetin attenuates KLF4-mediated phenotypic switch of VSMCs to macrophage-like cells in atherosclerosis: A critical role for the JAK2/STAT3 pathway

Journal: International Journal of Molecular Sciences

Response to Reviewers' comments

Dear Editor,

We express our gratitude for your diligent examination and meticulous evaluation of our manuscript. We value your insightful feedback and constructive initial response, which have guided our efforts to implement enhancements aimed at refining the manuscript. In response to the thorough scrutiny of the reviewers' comments, we have undertaken requisite revisions to amplify the lucidity and accessibility of the content, with the primary objective of facilitating enhanced reader comprehension.

We hope that you will find the revised manuscript suitable for publication. We eagerly anticipate the opportunity to contribute to the scholarly discourse within your publication. Please feel free to reach out to us should any further inquiries or concerns regarding the manuscript arise. Your continued engagement is greatly appreciated.

With kind regards,

Sincerely yours,

Lin Xiao

Tel: +86-15527269116, +86-0731-84487130

Xiangya School of Public Health, Central South University, Changsha, China

Reviewer #2

General comment: The scientific work presented demonstrates the interesting versatility of atherosclerosis and reveals a fundamental signaling pathway for the switch of VSMCS in Macrofagi-like. In genreale the work is well presented and designed, my advice is to minimize repetition between the various sections: introduction-discussion or materials and methods- results.

Response: We extend our sincere gratitude to Reviewer #2 for their scholarly review, insightful comments, and detailed feedback, all of which have been of great importance and have served as valuable guidance in enhancing the manuscript. We have revised the manuscript in accordance with your suggestions, and we provide comprehensive responses to each of the points you raised below.

Comment 1: I suggest to the authors to evaluate, in this work or in the future, how much this phenotypic switch and the attenuating action of KLF4 is mediated by the consilidata antioxidant action of quercetin, and therefore mediated also by other signaling or the action is preponderant on KLF4.

Response: Thank you very much for your constructive suggestions, which have served as a welcome guideline for our future research. Just as you mentioned, quercetin is a flavonoid derived from plants that has multiple biological activities inherently, including antioxidant properties[1]. By reviewing relevant literature, studies from Xi et al. [2] also demonstrated that the effect of quercetin protecting against the oxidative cell damage is mediated by inhibiting the expression of KLF4, and then attenuating the expressions of apoptotic target genes of KLF4 in neuroblastoma SH-SY5Y cells. Meanwhile, we have already discussed this in the “3. Discussion” section of our manuscript and marked with yellow. The revised content was showed as follows:

“Another research from Xi et al. [2] also demonstrated that quercetin alleviates the oxidative damage by suppressing the KLF4 expression in neuroblastoma SH-SY5Y cells.” (Page 16, Line 358-360).

However, it is noteworthy that KLF4, a member of the zinc-finger transcription factor family required for mammalian embryonic development, plays an important role in the cell proliferation and differentiation [3]. The JAK/STAT pathway, a well-known contributor of cell growth, is constitutively expressed in cultured VSMCs and responsible for VSMC proliferation [4]. Previous studies have reported that ox-LDL blocks VSMCs phenotypic switch by activating JAK2/STAT3 pathway [5,6]. Furthermore, Xue et al. [7] reported that the expression of p-STAT3 peaked in macrophage-like VSMCs to promote atherosclerosis in HFD-fed ApoE KO mice at the early stage. In the present study, we used various experimental methods along with different activators and inhibitors to comprehensively verify the reliability and scientificity of the results described in our manuscript. Therefore, there is no doubt that quercetin has antioxidant properties, but the positive effects of quercetin indicated in our study are still convincible and meaningful. We thank you again for your understanding.

  1. Kawabata, K.; Mukai, R.; Ishisaka, A. Quercetin and related polyphenols: new insights and implications for their bioactivity and bioavailability. Food Funct. 2015, 6, 1399-1417, doi:10.1039/c4fo01178c.
  2. Xi, J.; Zhang, B.; Luo, F.; Liu, J.; Yang, T. Quercetin protects neuroblastoma SH-SY5Y cells against oxidative stress by inhibiting expression of Kruppel-like factor 4. Neurosci. Lett. 2012, 527, 115-120, doi:10.1016/j.neulet.2012.08.082.
  3. McConnell, B.B.; Yang, V.W. Mammalian Kruppel-like factors in health and diseases. Physiol. Rev. 2010, 90, 1337-1381, doi:10.1152/physrev.00058.2009.
  4. Seki, Y.; Kai, H.; Shibata, R.; Nagata, T.; Yasukawa, H.; Yoshimura, A.; Imaizumi, T. Role of the JAK/STAT pathway in rat carotid artery remodeling after vascular injury. Circ.Res. 2000, 87, 12-18, doi:10.1161/01.res.87.1.12.
  5. Wang, T.; Xie, L.; Bi, H.; Li, Y.; Li, Y.; Zhao, J. Urantide alleviates the symptoms of atherosclerotic rats in vivo and in vitro models through the JAK2/STAT3 signaling pathway. Eur. J. Pharmacol. 2021, 902, 174037, doi: 10.1016/j.ejphar.2021.174037.
  6. Niwa, H.; Ogawa, K.; Shimosato, D.; Adachi, K. A parallel circuit of LIF signalling pathways maintains pluripotency of mouse ES cells. Nature 2009, 460, 118-122, doi:10.1038/nature08113.
  7. Xue, Y.; Luo, M.; Hu, X.; Li, X.; Shen, J.; Zhu, W.; Huang, L.; Hu, Y.; Guo, Y.; Liu, L. et al. Macrophages regulate vascular smooth muscle cell function during atherosclerosis progression through IL-1beta/STAT3 signaling. Commun. Biol. 2022, 5, 1316, doi:10.1038/s42003-022-04255-2.

Comment 2: In the introduction I suggest to better describe the macrophages-like phenotype and what characteristics/markers define it.

Response: Thank you very much for your constructive comments. We have modified the introduction to define the macrophage-like phenotype in detail, and the relevant information is now added in the manuscript where relevant (Page 2, Line 45-52).

“The accumulation of lipids within plaques progressively induces the phenotypic switch of VSMCs to macrophage-like cells, marked by the up-regulation of classic macrophage markers (e.g., CD68, F4/80 and LGALS3) and the down-regulation of VSMCs-specific markers like alpha smooth muscle actin (α-SMA) [1,2]. These VSMCs-derived macrophage-like cells exhibit heightened sensitivity to cholesterol levels and lipid loading, subsequently contributing to the formation of cholesterol-filled foam cells [3], emblematic pathological hallmarks of atherosclerosis.”

  1. Allahverdian, S.; Chehroudi, A.C.; McManus, B.M.; Abraham, T.; Francis, G.A. Contribution of intimal smooth muscle cells to cholesterol accumulation and macrophage-like cells in human atherosclerosis. Circulation 2014, 129, 1551-1559, doi:10.1161/CIRCULATIONAHA.113.005015.
  2. Rong, J.X.; Shapiro, M.; Trogan, E.; Fisher, E.A. Transdifferentiation of mouse aortic smooth muscle cells to a macrophage-like state after cholesterol loading. Proc. Natl. Acad. Sci. U. S. A. 2003, 100, 13531-13536, doi:10.1073/pnas.1735526100.
  3. Vengrenyuk, Y.; Nishi, H.; Long, X.; Ouimet, M.; Savji, N.; Martinez, F.O.; Cassella, C.P.; Moore, K.J.; Ramsey, S.A.; Miano, J.M. et al. Cholesterol loading reprograms the microRNA-143/145-myocardin axis to convert aortic smooth muscle cells to a dysfunctional macrophage-like phenotype. Arterioscler. Thromb. Vasc. Biol. 2015, 35, 535-546, doi:10.1161/ATVBAHA.114.304029.

Comment 3: in each section of the manuscript, please reduce the repetitions, concepts and details already described in other paragraphs (for example: line 124-128, 150-159 and many more).

Response: Thank you very much for your serious review and important advice. We have carefully reviewed our manuscript and thoroughly revised it to reduce duplication of concept and detail, particularly for the Materials and Methods-Results and Introduction-Discussion sections. The specific explanations for major revisions are listed below.

Firstly, we have merged the content previously mentioned in line 124-128 into “(E-H) Representative images and quantification of aortic cross-sections stained with H&E and Oil Red O staining” (Page 5, Line 120-122) by deleting duplicate information. Meanwhile, the content of Figure 2 has been revised to “Representative immunofluorescence images of aortic cross-sections to illustrate the co-localization of α-SMA (pink), CD68 (green) with KLF4, p-JAK2 and p-STAT3 (red). Nuclei were stained with DAPI (blue)” (Page 5, Line 146-148) by deleting duplicate content. In addition, we have carried out the same re-writing work on some of the other remaining sections, including “2. Results” (Page 7, Line 174-176; Page 8, Line 203-204; Page 11, Line 230-232; Page 13, Line 254-255; Page 15, Line 278-280) and “3. Discussion" (Page 15, Line 293-295, Line 303-305).

Comment 4: Results: Review the statistics so that the ranges of significance are consistent between the different graphs (for example: ** does not always correspond to all the differences between control and bars considered).

Response: Thank you very much for your scholarly review on our manuscript and kind reminding. We have carefully reviewed and seriously corrected all of the statistics to ensure consistency throughout the results. For example, we have revised Figure 3H's statistical chart as marked in yellow below (Page 6, Line 168). Thank you again for your valuable comments.

Figure 3. Ox-LDL treatment triggered phenotypic switch of VSMCs to macrophage-like cells in MOVAS at different concentrations. (A) Evaluation of MOVAS cell viability upon exposure to various concentrations of ox-LDL using the CCK8 assay, n = 6. (B) Representative images (Left, scale bar = 150 µm, 200×) and quantitative analysis (Right) of Oil Red O staining in ox-LDL-treated MOVAS. (C) Representative fluorescence images (Left, scale bar = 20 µm) and quantitative analysis (Right) of MOVAS treated with ox-LDL and labeled with Dil-ox-LDL. (D-H) Representative western blot and quantitative analysis for F4/80, CD68, α-SMA, LGALS3 and GAPDH proteins in MOVAS. P<0.05, ∗∗P <0.01 vs 0 µg/mL ox-LDL group.

Comment 5: It is unclear why, if the control in the figures has value 1, and then all values are normalized on them (using the control as a reference), there is a standard deviation on the control bar. Please clarify this point in the captions or in the materials and methods.

Response: Thank you very much for your constructive comments. Just as you concerned, we took the average of the three control group values before performing the lane normalization. Then, we divided each control group value by the average value to obtain the normalized values, which resulted in the bar values. Thank you again for your valuable feedback.

Comment 6: line 134-136: rewrite more clearly.

Response: Thank you very much for your constructive comments. We have revised the text for clarity. The revised sentence now reads as follows: "...TSA multi-color immunofluorescence staining was employed to examine the co-localization of macrophage-like VSMCs, identified as double-positive for α-SMA (pink) and CD68 (green), with KLF4, p-JAK2, and p-STAT3 (red) in the aortic tissues of ApoE KO mice, respectively.” (Page 5, Line 134-135).

Comment 7: Arrows may help in the images to indicate the highlights to look at, especially in IF on fabric.

Response: Thank you very much for your valuable comments. We appreciate your attention to detail. We have added arrows to highlight the key features in the IF images on fabric and have incorporated these changes in the revised manuscript, as shown in Figure 2 (Page 5, Line 145).

Figure 2. Representative immunofluorescence images of aortic cross-sections to illustrate the co-localization of α-SMA (pink), CD68 (green) with KLF4, p-JAK2 and p-STAT3 (red). Nuclei were stained with DAPI (blue). Scale bar =100 μm, 100 ×; scale bar = 20 μm, 400 ×; n = 3.

Comment 8: Figures: In general, I suggest that authors increase the magnification or intensity of the Dil-ox-LDL images so that it is more visible and appreciable by the reader. For Oil Red O figures, I suggest increasing the contrast so that the differences are more visible.

Response: Thank you for your valuable feedback regarding the figures in our manuscript. We have adjusted the intensity of the Dil-ox-LDL images to ensure that the details are clearer and more discernible. Additionally, we have increased the contrast to enhance the visibility of Oil Red O figures, making the distinctions more prominent. We have updated the manuscript with these improved images as marked in yellow below, including Figure 5-7 (Page 10, Line 224; Page 12, Line 249; Page 14, Line 273).

Figure 5. Quercetin blocked phenotypic switch of VSMCs to macrophage-like cells and foam cell formation through reducing KLF4 expression. (A) Representative western blot for KLF4 protein in MOVAS. (B) Quantitative analysis for the relative protein expression of KLF4 in MOVAS. (C) Representative images (Left, scale bar = 150 µm, 200×) and quantitative analysis (Right) of Oil Red O staining in MOVAS. (D) Representative fluorescence images (Left, scale bar = 20 µm) and quantitative analysis (Right) of MOVAS labeled with Dil-ox-LDL. (E-J) Representative western blot and quantitative analysis for F4/80, KLF4, CD68, α-SMA, LGALS3 and GAPDH protein in MOVAS P<0.05, ∗∗P<0.01 vs control group; #P<0.05, ##P<0.01 vs ox-LDL group; &P<0.05, &&P<0.01 vs ox-LDL + Que group.

Figure 6. Ox-LDL induced phenotypic switch of VSMCs to macrophage-like cells in MOVAS by activating JAK2/STAT3 pathway. (A-B) Representative images (A) (scale bar = 150 µm, 200×) and quantitative analysis (B) of Oil Red O staining in MOVAS. (C-D) Representative fluorescence images (C) (scale bar = 20 µm) and quantitative analysis (D) of MOVAS labeled with Dil-ox-LDL. (E-L) Representative western blot and quantitative analysis for p-JAK2, JAK2, p-STAT3, STAT3, F4/80, KLF4, CD68, α-SMA, LGALS3 and GAPDH protein in MOVAS. P<0.05, ∗∗P<0.01 vs control group; #P<0.05, ##P<0.01 vs ox-LDL group.

Figure 7. Quercetin improved the ox-LDL-induced phenotypic switch of VSMCs to macrophage-like cells in MOVAS by reduction of the activated JAK2/STAT3 pathway. (A) Representative images (Left, scale bar = 150 µm, 200×) and quantitative analysis (Right) of Oil Red O staining in MOVAS. (B) Representative fluorescence images (Left, scale bar = 20 µm) and quantitative analysis (Right) of MOVAS labeled with Dil-ox-LDL. (C-J) Representative western blot and quantitative analysis for p-JAK2, JAK2, p-STAT3, STAT3, F4/80, KLF4, CD68, α-SMA, LGALS3 and GAPDH protein in MOVAS. P<0.05, ∗∗P<0.01.

Comment 9: In the panels containing the blots, I recommend replacing the blots too impressed and over exposed, in fact too intense and coarse bands interfere with the quantification, also preventing to clearly observe the modulation of the analyzed markers.

Response: Thank you for your valuable feedback on our manuscript. You correctly pointed out that some of the Western blot images were overexposed and showed excessively intense bands, which could interfere with quantification and obscure the modulation of the analyzed markers. We have taken reassemble the original strips to replace the blots too impressed and over exposed in Figure 6E and Figure 7C (Page 12, Line 249; Page 14, Line 273), ensuring that the resulting images are clearer and more accurate.

Figure 6. Ox-LDL induced phenotypic switch of VSMCs to macrophage-like cells in MOVAS by activating JAK2/STAT3 pathway. (A-B) Representative images (A) (scale bar = 150 µm, 200×) and quantitative analysis (B) of Oil Red O staining in MOVAS. (C-D) Representative fluorescence images (C) (scale bar = 20 µm) and quantitative analysis (D) of MOVAS labeled with Dil-ox-LDL. (E-L) Representative western blot and quantitative analysis for p-JAK2, JAK2, p-STAT3, STAT3, F4/80, KLF4, CD68, α-SMA, LGALS3 and GAPDH protein in MOVAS. P<0.05, ∗∗P<0.01 vs control group; #P<0.05, ##P<0.01 vs ox-LDL group.

Figure 7. Quercetin improved the ox-LDL-induced phenotypic switch of VSMCs to macrophage-like cells in MOVAS by reduction of the activated JAK2/STAT3 pathway. (A) Representative images (Left, scale bar = 150 µm, 200×) and quantitative analysis (Right) of Oil Red O staining in MOVAS. (B) Representative fluorescence images (Left, scale bar = 20 µm) and quantitative analysis (Right) of MOVAS labeled with Dil-ox-LDL. (C-J) Representative western blot and quantitative analysis for p-JAK2, JAK2, p-STAT3, STAT3, F4/80, KLF4, CD68, α-SMA, LGALS3 and GAPDH protein in MOVAS. P<0.05, ∗∗P<0.01.

Comment 10: Line 189-190: "strongly reversed" is not observable by the blots presented. Please clarify this discrepancy.

Response: Thank you very much for your constructive comments. As shown in Figure 4D-H, ox-LDL increased the expression of F4/80, CD68, and LGALS3 levels, while decreased the levels of α-SMA (all P<0.05). Compared to the ox-LDL group, ox-LDL + quercetin group markedly reversed the levels of F4/80, CD68, and LGALS3, and counteracted the ox-LDL-induced reduction in α-SMA levels. We have modified the statement in the text to describe the results clearly and unambiguously as follows.

“Moreover, compared to the ox-LDL group, quercetin supplementation markedly reversed the ox-LDL-induced levels of F4/80, CD68, and LGALS3, and counteracted the ox-LDL-induced reduction in α-SMA levels ....” (Page 7, Line 188-190).

Comment 11: Discussion: The discussion is well written, my concern is about the repetition of concepts that overlap with the introduction.

Response: Thank you for your valuable comments on our discussion section. In response to your concern about the repetition of concepts in the discussion overlapping with the introduction, we have now reduced the repetitive descriptions of quercetin and phenotypic switch of VSMCs to macrophage-like cells in the discussion section. We have also used more appropriate and concise language to describe these concepts. We believe these revisions will improve the overall quality of our manuscript. Thank you again for your valuable comments. The relevant information is in the Discussion section and marked with yellow as follows.

“An increasing number of studies have demonstrated the protective effects of quercetin against atherosclerosis by regulating lipid levels and preventing TC accumulation [1-3]. ” (Page 15, Line 293-295).

“There is increasing evidence that macrophage-like VSMCs promote the development of atherosclerosis by contributing to the formation of cholesterol-filled foam cells [4,5].” (Page 15, Line 303-305).

  1. Dalgaard, F.; Bondonno, N.P.; Murray, K.; Bondonno, C.P.; Lewis, J.R.; Croft, K.D.; Kyro, C.; Gislason, G.; Scalbert, A.; Cassidy, A. et al. Associations between habitual flavonoid intake and hospital admissions for atherosclerotic cardiovascular disease: a prospective cohort study. Lancet Planet. Health 2019, 3, e450-e459, doi:10.1016/S2542-5196(19)30212-8.
  2. Xiao, L.; Liu, L.; Guo, X.; Zhang, S.; Wang, J.; Zhou, F.; Liu, L.; Tang, Y.; Yao, P. Quercetin attenuates high fat diet-induced atherosclerosis in apolipoprotein E knockout mice: A critical role of NADPH oxidase. Food Chem. Toxicol. 2017, 105, 22-33, doi: 10.1016/j.fct.2017.03.048.
  3. Luo, G.; Xiang, L.; Xiao, L. Quercetin alleviates atherosclerosis by suppressing oxidized LDL-induced senescence in plaque macrophage via inhibiting the p38MAPK/p16 pathway. J. Nutr. Biochem. 2023, 116, 109314, doi: 10.1016/j.jnutbio.2023.109314.
  4. Vengrenyuk, Y.; Nishi, H.; Long, X.; Ouimet, M.; Savji, N.; Martinez, F.O.; Cassella, C.P.; Moore, K.J.; Ramsey, S.A.; Miano, J.M. et al. Cholesterol loading reprograms the microRNA-143/145-myocardin axis to convert aortic smooth muscle cells to a dysfunctional macrophage-like phenotype. Arterioscler. Thromb. Vasc. Biol. 2015, 35, 535-546, doi:10.1161/ATVBAHA.114.304029.
  5. Tieyuan, Z.; Ying, Z.; Xinghua, Z.; Huimin, W.; Huagang, L. Piceatannol-mediated JAK2/STAT3 signaling pathway inhibition contributes to the alleviation of oxidative injury and collagen synthesis during pulmonary fibrosis. Int. Immunopharmacol. 2022, 111, 109107, doi: 10.1016/j.intimp.2022.109107.

Comment 12: Materials and methods: Please enter the stimulation times of in vitro experiments to evaluate the protein expression of the presented markers.

Response: Thank you very much for your careful review. The stimulation times of in vitro experiments has been described in the Materials and Methods section “4.5. MOVAS culture” as follows: “After 24 hours, cells were used for subsequent experimentals” (Page 18, Line 435-436). We thank you again for your thorough review.

Round 2

Reviewer 1 Report

Comments and Suggestions for Authors

The authors did not address most of my previous review comments.  This reviewer cannot say the manuscript has been improved.  

Comments on the Quality of English Language

Minor editing of English language required

Round 3

Reviewer 1 Report

Comments and Suggestions for Authors

My concern is about the quality of the immunofluorescence studies.  For example, SM a-actin was stained only in the lower portion of vessels in normal diet group (Figure 2 left, middle panel).    In spite of this, a magnified image was taken from the upper portion of this picture (i.e., SM a-actin negative area).   As such, the images should be improved.   

Comments on the Quality of English Language

Minor editing of English language required

Author Response

Comments 1:My concern is about the quality of the immunofluorescence studies. For example, SM a-actin was stained only in the lower portion of vessels in normal diet group (Figure 2 left, middle panel). In spite of this, a magnified image was taken from the upper portion of this picture (i.e., SM a-actin negative area). As such, the images should be improved.

Response:

Thank you very much for your careful review on our manuscript and kind reminding. While we acknowledge your reservations regarding the quality of our immunofluorescence results, it is imperative to emphasize that our images are devoid of any quality deficiencies. Based on your description, it appears that the predominance of α-SMA staining in the lower portion of vessels in the normal diet group (left panel, middle section of Figure 2) may have been misinterpreted as indicative of image quality concerns, especially given the absence of α-SMA staining in the upper portion (specifically the region demarcated by green lines, Figure R1). However, upon further analysis of the unaltered 100x magnified immunofluorescence image of this picture presented in Figure R1, it is apparent that there is clear α-SMA staining within the delineated red boxed area. Additionally, the proximity of the red region 2 to the green-marked area indicates a consistent presence of α-SMA staining in the upper portion.

The lack of α-SMA staining in the green region is likely due to the anatomical curvature of the mouse aortic sinus, which includes three valves. When dissecting this structure, it is possible that not all valves are uniformly aligned in the same plane, leading to incomplete sections of smooth muscle tissue in certain valves. This phenomenon may account for the partial staining of α-SMA in only a subset of valves, a common observation in such cases. After conducting a thorough review of the existing literature, it was observed that there is a scarcity of studies that provide fluorescence staining results for all three aortic valves[1, 2], a gap that our study aims to fill. The majority of studies that do include such results demonstrate partial staining of α-SMA in aortic valves, as exemplified by Aryal et al. [3] and Zotes et al. [4] in Figure R2 and Figure R3, respectively. Therefore, our fluorescence findings should not be attributed to a lack of quality. While α-SMA staining may not always be present in aortic valves, the identification of α-SMA+CD68+ double-positive cells within the plaque suggests a potential macrophage phenotypic switch.

References:

  • Delgado-Maroto V, Benitez R, Forte-Lago I, Morell M, Maganto-Garcia E, Souza-Moreira L, O'Valle F, Duran-Prado M, Lichtman AH, Gonzalez-Rey E, Delgado M. Cortistatin reduces atherosclerosis in hyperlipidemic ApoE-deficient mice and the formation of foam cells. Sci Rep. 2017 Apr 13;7:46444. PMID: 28406244
  • Furusho Y, Miyata M, Matsuyama T, Nagai T, Li H, Akasaki Y, Hamada N, Miyauchi T, Ikeda Y, Shirasawa T, Ide K, Tei C. Novel Therapy for Atherosclerosis Using Recombinant Immunotoxin Against Folate Receptor β-Expressing Macrophages. J Am Heart Assoc. 2012 Aug;1(4):e003079.PMID: 23130174.
  • Aryal B, Rotllan N, Araldi E, Ramírez CM, He S, Chousterman BG, Fenn AM, Wanschel A, Madrigal-Matute J, Warrier N, Martín-Ventura JL, Swirski FK, Suárez Y, Fernández-Hernando C. ANGPTL4 deficiency in haematopoietic cells promotes monocyte expansion and atherosclerosis progression. Nat Commun. 2016 Jul 27;7:12313. PMID: 27460411.
  • Zotes TM, Arias CF, Fuster JJ, Spada R, Pérez-Yagüe S, Hirsch E, Wymann M, Carrera AC, Andrés V, Barber DF. PI3K p110γ deletion attenuates murine atherosclerosis by reducing macrophage proliferation but not polarization or apoptosis in lesions. PLoS One. 2013 Aug 22;8(8):e72674. PMID: 23991137.

Round 4

Reviewer 1 Report

Comments and Suggestions for Authors

My concerns are cleared.